# Therapeutic Perspectives for Microbiota Transplantation in Digestive Diseases and Neoplasia—A Literature Review

**DOI:** 10.3390/pathogens12060766

**Published:** 2023-05-26

**Authors:** Adrian Boicean, Dan Bratu, Ciprian Bacila, Ciprian Tanasescu, Radu Sorin Fleacă, Calin Ilie Mohor, Andra Comaniciu, Teodora Băluță, Mihai Dan Roman, Radu Chicea, Adrian Nicolae Cristian, Adrian Hasegan, Sabrina Birsan, Horațiu Dura, Cosmin Ioan Mohor

**Affiliations:** 1County Clinical Emergency Hospital of Sibiu, 550245 Sibiu, Romania; adrian.boicean@ulbsibiu.ro (A.B.); ciprian.bacila@ulbsibiu.ro (C.B.); ciprian.tanasescu@ulbsibiu.ro (C.T.); radu.fleaca@ulbsibiu.ro (R.S.F.); calin.mohor@ulbsibiu.ro (C.I.M.); comaniciuandra@gmail.com (A.C.); olteanteodora@yahoo.com (T.B.); mihai.roman@ulbsibiu.ro (M.D.R.); radu.chicea@ulbsibiu.ro (R.C.); adrian.cristian@ulbsibiu.ro (A.N.C.); adrian.hasegan@ulbsibiu.ro (A.H.); sabrinaandreea.marinca@ulbsibiu.ro (S.B.); horatiu.dura@ulbsibiu.ro (H.D.); cosmin.mohor@ulbsibiu.ro (C.I.M.); 2Faculty of Medicine, Lucian Blaga University of Sibiu, 550169 Sibiu, Romania

**Keywords:** fecal microbiota transplantation, Clostridium difficile, extra-digestive diseases, melanoma, cancer

## Abstract

In a mutually beneficial connection with its host, the gut microbiota affects the host’s nutrition, immunity, and metabolism. An increasing number of studies have shown links between certain types of disease and gut dysbiosis or specific microorganisms. Fecal microbiota transplantation (FMT) is strongly advised for the treatment of recurrent or resistant Clostridium difficile infection (CDI) due to its outstanding clinical effectiveness against CDI. The therapeutic potential of FMT for other disorders, particularly inflammatory bowel diseases and malignancies, is currently gaining more and more attention. We summarized the most recent preclinical and clinical evidence to show the promise of FMT in the management of cancer as well as complications related to cancer treatment after reviewing the most recent research on the gut microbiota and its relationship to cancer.

## 1. Introduction

The gut microbiota, imprinted in our genes, is also influenced by a number of various triggers, such as food, environment, and many other factors. Genetic mapping of the gut microbiota established that it contains 100 billion microorganisms on average, which is 10 times more than the human body’s cells. It is important to outline that the phyla *Firmicutes* and *Bacteroidetes* make up the majority of the healthy gut microbiota, followed by the phyla *Actinobacteria* and *Verrucomicrobia*; dysbiosis may change the number of saprophytes microbes and cause different diseases [1]. Environmental factors might cause differences in gut microbiota at various times in the same anatomical location in the same person [2,3]. There will be a noticeable variation in the diversity and quantity of bacteria from the esophagus to the rectum due to genetic predisposition, alimentation triggers, and other environmental factors that change the microbiota and cause dysbiosis [3,4,5]. Moreover, the consumption of antibiotics is another gate toward dysbiosis due to so many factors that change the balance between healthy and unhealthy bacteria; FMT is a significant treatment in improving gut health [1]. The gastrointestinal microbiota and their genetic products play a significant role in the development of the immune system, host defense, energy metabolism, and nutrition in a complicated yet balanced equilibrium. The improper compositional perturbation of this balance, known as dysbiosis, has been linked to a number of illness conditions [2]. Many factors influence the normal microbial composition of the intestine; medication, particularly broad-spectrum antibiotics, is by far the most common factor [2,3].

The intestinal microbiota is involved in the response to various diseases by influencing different metabolic axes and the immune system. Recent studies have shown that the human microbiota is also involved in the immune response to SARS-CoV-2 infection. Many recent studies suggest that SARS-CoV-2 leads to changes in the intestinal microbiota by affecting ACE2, mitochondrial damage, and changes in the microbiota–lung axis or microbiota liver–gut axis, a concept which is highlighted by new studies [4,6,7]. It is also possible that intestinal inflammation due to COVID-19 could predispose individuals to other gastrointestinal pathogens. Furthermore, many disease complications have a positive correlation with an overpopulation of pathogenic bacteria (*Coprobacillus*, *Clostridium ramosum*, *Clostridium hathewayi*). A decrease in the beneficial anti-inflammatory bacteria *Faecalibacterium prausnitzll*, *phyla Firmicutes*, and *Bacteroidetes*, which represent the majority of the healthy gut microbiota, the phyla Actinobacteria and *Verrucomicrobia* are important in the gut homeostasis [4,8,9].

Recent studies and case reports focus on the importance of gut microbiota in oncogenetic development and response to chemotherapy in various types of cancer [10,11]. Additionally, a recent review realized by Yuanyuan Zhao et al. outlines the implications of the gut microbiota in developing liver diseases such as fatty liver disease, cirrhosis, and even implications in hepatocarcinoma. This review highlights the importance of FMT by assessing the gut–liver axis in recovering from these pathologies [4,12,13,14,15]. Since specialized studies have demonstrated a change in the intestinal microbiota in patients with rheumatoid arthritis, with the frequent presence of *Bacteroides*, *Escherichia–Shigella*, and *Prevotella* bacteria, at Chinese PLA General Hospital, FMT was successful in a 20-year-old patient with a 5-year history of rheumatoid arthritis [16].

In order to directly modify the recipient’s microbial composition and provide a health benefit, fecal microbiota transplantation (FMT) involves administering a solution of fecal material from a donor into a recipient’s digestive tract [5,6,17]. Because of its clinical benefit of changing the gut microbiota of its receivers, FMT is regarded as the most cutting-edge approach to treating dysbiosis and its complications [4,5,18]. According to 2013 guidelines, FMT has been approved as a therapeutic treatment for treating recurrent Clostridium difficile infection (CDI), and its clinical effectiveness has reached about 92% [6,8,18,19]. FMT has so far been a successful therapy method for recurrent CDI. However, it is anticipated that FMT standards will be established in the upcoming years in addition to conventional methods and that its regulations will expand [18].

In pediatric patients with recurrent CDI, the safety and effectiveness of FMT can be ensured by a thorough analysis of the effect on the gut microbiota [20]. A series of studies show the beneficial effects of fecal transplantation in children [21,22], but the technical aspects (quantity, delivery method, donor screening protocols, etc.) require more attention [22]. Frequently, in children with co-morbidities, FMT did not present positive results [23].

The first outcome of this review is to investigate the evidence surrounding the use of FMT in CDI and other pathologies that outline the clinical benefits of FMT. We focused especially on the oncological pathology and adverse reactions given by chemotherapy and immunotherapy of these diseases, inflammatory bowel diseases, but considering the COVID-19 pandemic, we also considered some aspects related to digestive problems caused by the SARS-CoV-2 virus. Through this review, we aim to encourage practitioners to consider the clinical benefits of FMT in the case of CDI but also in other digestive and extra-digestive pathologies. We outlined multiple ways of administration, from crude fresh stool to new perspectives such as washed microbiota and spore therapy to liquid capsules.

## 2. Materials and Methods

According to the search engines Google Scholar, Web of Science, PubMed, Medscape, and UpToDate, in the period January 2018–July 2022, over 17,000 specialized studies were found with reference to treatment with FMT. We approached the most recent analyses, studies, and research related to fecal microbiota transplantation. We noted the following: FMT programs in developing countries; systematic reviews and meta-analyses; fecal microbiota transplantation in children; and the use of FMT in the treatment of other diseases. Below is a list of alternatives used for searching databases: “fecal”, “faecal”, “microbiota”, “microflora”, “feces”, “faeces”, and “stool”. These were searched individually or with the following options of transplant lexical field: “transplant”, “implant”, “donor”, “donation”, “enema”, “reconstitution”, “infusion”, “therapy”, and “bacteriotherapy”. In order to ensure a complete search result, all of these terms were searched independently as well as in combination with various descriptive terms such as “FMT in melanoma”, “*Clostridoides difficile*”, “Healthy gut microbiota”, “cancer”, and “melanoma”. In the last period, the specialized literature offers multiple studies and research aimed to affirm positive results for fecal microbiota transplantation in different conditions.

## 3. Feces Preparation, Donor Screening, and Safety in FMT

Studies show that usually, first or second-degree relatives are selected as potential donors according to the current clinical guidelines. A stool bank may be used in order to improve the safety of the method. The fecal matter samples were tested copro-parasitologically and virologically (Ag HBS, AcVHC, HIV, CMV, CD), as well as for COVID-19. Donors were tested for SARS-CoV-2 (by RT-PCR test) [6,8,24].

Exclusion criteria were patients with documented autoimmune or other chronic diseases, patients that had undergone major surgery or had received blood products in the 12 months previous to fecal matter collection were also excluded, as well as donors over the age of 50. Studies show that the eligibility for each donor is based on their medical records and a personal interview [6,8]. The day before FMT, the patients were prepared by intestinal lavage with 4 L of PEG (polyethylene glycol). The fecal matter transfer technique consisted of 50 to 90 g of donated feces (less than 6 hours after defecation for fresh stool donation), also frozen stool seems to be as effective as fresh feces, dissolved in 500 mL of saline 0.9%, then mixed to obtain a homogeneous solution and filtered. Subsequently, a total colonoscopy or enema may be performed without sedation in order to preserve anal sphincter control. Additionally, FMT via the gastro-jejunal route is described in studies, frequently in cases of Crohn’s disease with gastro-duodenal lesions. However, we have noted that the majority of studies describe positive results in the case of colonoscopy or enema [6,8,25].

Very few adverse effects are directly attributed to the procedure. Most reported adverse effects have been self-limiting gastrointestinal symptoms such as abdominal cramps, nausea, and constipation. According to studies, very rare adverse effects are fever, Gram-negative bacteremia, and bowel perforation [8,26].

Current perspectives on FMT involve washed microbiota transplantation which represents an innovative way to improve safety and lower potential risks. Laboratory preparation of the washed microbiota in comparison to fresh/frozen microbiota transplantation involves microfiltration, which consists of an automatic purification system and then repeated centrifugation plus three-time suspension in order to purify the feces. The study realized by Ting Zhang et al. in comparing washed microbiota transplantation (WMT) with manual transplantation noted that the microfiltration system improved safety in WMT compared to crude feces in cases of UC (ulcerative colitis) and CD (Crohn’s disease), reporting lower numbers of adverse effects and bacteriemia [19,27]. For patients that require repeated FMT or present a high risk for perforation, a new technique for delivery has been developed: colonic transendoscopic enteral tubing (TET), which is a new way for instillation of FMT that reduces the risk of perforation [27]. In the study of Lu et al., they outlined that colonic TET could be considered the most effective way of delivery of WMT in patients with ulcerative colitis, which reduces possible perforation risks [28].

Gweon et al. describe the next step in fecal microbiota transplantation based on spores therapy that involves sequencing techniques that help analyze the gut particularities of the patient and the donor. The spore formulation is based on centrifugation with ethanol and then capsulated, and studies show that this type of therapy improves dysbiosis and prevents recurrent CDI. However, the dosage and time of administration require further studies to improve this medical device [25,29,30,31,32].

Other effective stool formulations are represented by liquid capsules that infuse frozen or fresh stool into capsules, lyophilized capsules that are dehydrated, and powdered feces, and that also showed 78%–87.5% efficacy in CDI and were as effective as fresh/frozen microbiota transplantation [29]. Youngster et al. concluded in their study on the safety and preliminary efficacy of orally administered lyophilized fecal microbiota product compared with frozen product given by enema that both oral or rectal administration of lyophilized fecal microbiota transplantation showed equivalent efficacy; however, they noted that the dose should be higher in lyophilized capsules [29,30].

Furthermore, Kao et al. noted in their study that frozen FMT is as efficient as fresh FMT in treating recurrent CDI. Additionally, with regard to oral capsules, the study noted the same efficiency as FMT delivered by colonoscopy [31].

These perspectives and new formulas to deliver FMT enhance the convenience for practitioners to use FMT in various diseases, reduce adverse effects, and improve safety in CDI and non-CDI diseases. Alternative routes and ways of FMT delivery will also improve the success rate and clinical benefits in other diseases, such as inflammatory bowel diseases, oncological pathologies, and other autoimmune diseases [31,32].

## 4. Studies on FMT during the COVID-19 Pandemic

Positive perception of FMT, both for medical professionals and patients, can be promoted by applying the nudge principle (the main key of behavioral economics), which involves changing people’s behavior in order to make the best decision [5,17]. The increased interest in FMT has led specialists to carry out multiple studies to indicate the safety and effectiveness of this therapeutic operation. Since the intestinal microbiota plays a particularly important role in maintaining human health, FMT has been proven to be beneficial for gastrointestinal and extra-intestinal pathologies [33,34], with a rate of serious adverse events in less than 1% of CDI patients [35] and with a high success rate in patients with severe forms of CDI [6,36].

Lately, studies demonstrated positive results in patients with CDI and COVID-19 co-infection, lowering the inflammatory syndrome and reducing abdominal cramps and the rate of recurrence. According to the European Society of Clinical Microbiology and Infectious Diseases 2021 update on the treatment guidance document for *Clostridioides difficile* infection in adults, FMT is an approved treatment for recurrent or refractory CDI. However, the benefits of FMT on the gut microbiota and inflammation have opened new perspectives for this way of gut repopulation. Other diseases treated with FMT outlined the clinical potential of this treatment. There was concern about the COVID-19 pandemic’s impact on CDI frequencies and severity; while various remedies were being researched in the early stages of the pandemic, antibiotics were widely used, increasing the risk of CDI, particularly in hospitalized patients [7,37,38].

More than 60% of COVID-19 patients had gastrointestinal signs, and gastrointestinal symptoms such as diarrhea, nausea, and vomiting have been observed in previous investigations. The diarrhea symptom was present in all of the COVID-19 patient groups evaluated [6,7,37]. In the context of COVID-19, elucidating changes to the microbiota as valid biomarkers is an ignored piece of the disease puzzle that demands further exploration [39].

One study highlights the beneficial effects of FMT in patients with COVID-19 and CDI and the importance of repopulating the gut microbiome in order to restore immunological function, reduce systemic inflammation, and lower the chance of recurrence. More than 91% of FMT patients experienced no abdominal pain after therapy and a significant improvement in inflammatory syndrome [6].

A severe SARS-CoV-2 infection is accompanied by an increase in pro-inflammatory cytokines or the "cytokine storm", which unmistakably illustrates an unchecked dysregulation of the host’s immune system. Multiple organ failure and severe acute respiratory syndrome in the lungs can result from this increased cytokine and chemokine production. In addition to influencing the innate immune response, the gut microbiota also enhances CD8+ T cell effector function, which is a step in the viral (influenza) clearance process [40].

Biliński et al. describe the rapid resolution of COVID-19 after fecal microbiota transplantation in a case series of patients with improvement in inflammatory syndrome and alleviating of respiratory symptoms after FMT delivery [41].

We want to highlight the clinical benefits that FMT proved in the case of CDI and SARS-CoV-2, lowering the abdominal cramps, the inflammatory syndrome and lowering the recurrence rate in patients with CDI and SARS-CoV-2 co-infection.

## 5. FMT Clinical Benefits in Inflammatory Bowel Diseases

Despite receiving successful IBD treatment that has led to mucosal repair and disease remission, patients with an established diagnosis of IBD, such as those with Crohn’s disease (CD) or ulcerative colitis (UC), may nevertheless experience persistent gastrointestinal symptoms. Functional gastrointestinal symptoms are linked to anxiety, depression, poorer quality of life, and higher healthcare costs in IBD patients. As a result, it is critical to recognize functional gastrointestinal symptoms in this situation in order to select an efficient therapy strategy [42].

According to various scientific investigations, the modification of the gut microbiota composition (dysbiosis) is the primary stakeholder related to IBD pathogenesis. Unfortunately, the specific gut microbiota core composition and metabolic indicators that are thought to be important in the initiation of IBD pathogenesis remain unknown. Many animal studies have been conducted to evaluate FMT as a potential therapy option for a variety of gastrointestinal and other metabolic disorders. A variety of data suggest an important role of gut microbiota in IBD. FMT has recently received a lot of interest as a new therapeutic strategy in IBD [42,43,44].

FMT can be seen as a promising method for the treatment if the microbiota composition is significantly altered in IBD. In patients with UC and CD, preliminary clinical reports of FMT showed clinical remission that was maintained over a long period of follow-up in many cases and in a small number of additional cases also documented endoscopic and histologic remission [25,45,46,47]. A 36.2% remission rate was discovered in a recent review and meta-analysis of nine studies that included 122 patients who received FMT (79 with UC, 39 with CD, and 4 with unclassified IBD). In contrast to UC patients, where only 22% of patients obtained remission, the rate of remission was higher in younger patients (7–20 years old) and in individuals with CD (64.1 and 60.5%, respectively) [8,42,43,44].

A double-blinded randomized control study in Vermont, USA, aimed to evaluate the viability and safety of performing induction FMT via initial colonoscopy infusion followed by 12 weeks of ambulatory oral maintenance therapy with frozen FMT capsules. Participants in the study were required to have a confirmed diagnosis of UC, with inflammation reaching proximally to at least the recto-sigmoid junction. In the study, 15 subjects were recruited, from which 7 subjects were randomly assigned to the FMT and 8 to the placebo group. Three patients were eliminated from the trial because they did not match the endoscopic criteria for inclusion (Mayo score > 1). The remaining 12 participants (6 in each group) received at least one trial treatment dosage. One patient in the placebo group dropped out after 6 weeks due to disease progression. Two subjects that received frozen FMT capsules achieved clinical remission compared to none in the placebo group. These preliminary findings show that daily encapsulated frozen FMT may increase the duration of index FMT-induced alterations in gut bacterial community composition. Oral frozen encapsulated FMT is a promising FMT delivery system and may be preferred for long-term treatment strategies in UC and other chronic diseases, but further evaluations will have to address home storage concerns [47].

Mańkowska-Wierzbicka et al. published a pilot study in 2020 to evaluate the efficacy of multi-session FMT (weeks 1–6) treatment in active UC patients. Ten UC patients received multi-session FMT (200 mL) from healthy donors by colonoscopy/gastroscopy. Stool sample analyses were performed after the sixth session of FMT administration (week 7) and at the conclusion of the follow-up period (6 months) to examine changes in the microbiome and the concentration of fecal calprotectin. After 6 months, the diversity and richness of the recipients’ fecal microbiota increased but then declined significantly. The trial found that six rounds of rigorous, weekly FMT treatment improved clinical (Truelove and Witts score) and biochemical (CRP, calprotectin) outcomes not just immediately after FMT but also up to six months later. The success of this trial appears to be connected to the good donor microbial features as well as the intended scheme and route of FMT (one colonoscopy and five rounds of gastroduodenoscopy). The substantial proportion of patients (60%) who improved clinically underlines the usefulness of FMT in long-term microbial diversity modification in UC patients [46].

A pilot study was carried out in Italy in which two pediatric UC patients were discussed. They had both been treated with a number of cycles of steroids and antibiotics during the year prior to FMT. They both had a relapsing disease with recurrent episodes of bloody diarrhea and abdominal pain. Patient 1 was receiving mesalamine maintenance treatment. Left-sided colitis was discovered by endoscopic evaluation at the time of FMT with a Mayo score of 0 in the ascending, transverse, descending colon and 2 in the sigmoid–rectum. The FMT follow-up went without a hitch, and the patient reported no symptoms. Colonoscopy results after 12 weeks showed a decrease in disease activity with a Mayo score of 0 in the ascending, transverse, descending colon and 1 in the sigmoid–rectum. Mesalamine and azathioprine were being used in maintenance therapy for patient 2. At the time of FMT, an endoscopic evaluation identified a pancolitis with a Mayo score of 1 in the ascending, transverse colon and 2 in the descending colon–sigmoid–rectum. After FMT, there were no complications, and the patient reported clinical improvement and a decrease in bowel motions [25].

Kunde et al. presented a study with 10 pediatric patients that received enemas for five days; patients achieved remission after one week of fecal microbiota transplantation by enemas. This study shows the importance of FMT administration, even by enemas, without the need for a total colonoscopy [45].

CD with fistula and formation of an intraperitoneal big inflammatory mass has significant morbidity and remains an unresolved challenge. Zhang et al. presented a case of refractory CD complicated by fistula, residual Barium sulfate, and the formation of an intraperitoneal big inflammatory mass that was successfully treated with standardized FMT as a rescue therapy. This case report brings new perspectives for patients with fistulas to prevent surgery [48].

Another study described 133 subjects who received FMT: 77 of the patients suffered from UC, and 56 patients from CD, and the majority of the patients were resistant to therapy or dependent corticotherapy. A total of 57 of the patients evaluated in the study were also associated with recurrent CDI. The results of the study showed that FMT obtained a decrease of 71% in symptoms of the treated patients and also prevented relapsing of CDI infection [49] (Table 1).

This treatment approach should still be viewed as experimental because the FMT results in IBD patients are variable. More research is required on choosing eligible donors, picking highly responsive patients, and processing feces in anaerobic environments. Furthermore, we are yet unsure of the ideal timing for FMT therapies in IBD patients. Should FMT be utilized as the main course of treatment or as a post-induction measure? The good news is that a number of ongoing trials may contribute to answering the problems raised above. Furthermore, this will make it possible for FMT to be used as a potential therapy option for IBD patients in the future [50].

## 6. FMT Improving Treatment Response in Melanoma

Several studies have been conducted to investigate a possible link between gut microbiota and clinical response to cancer immunotherapy, particularly immune checkpoint inhibitors (ICPI) [51].

In a study published in 2019 by Youngster and colleagues, five patients with treatment-resistant metastatic melanoma were recruited. Two patients with advanced melanoma who experienced a long-lasting, full response to treatment served as the FMT donors. FMT was performed by both oral administration and colonoscopy, and then anti-PD-1 retreatment was used; each patient received complete body imaging, bowel and tumor tissue biopsies, and stool samples before and after treatment. Increased post-FMT infiltration of antigen-presenting cells CD68+ in the gut and tumor was seen by immunohistochemical labeling of samples. Following treatment, there was also an increase in CD8+ T cell infiltration. Following FMT, three patients experienced a full or partial response to treatment [52].

Sivan et al. describe the clinical benefits of commensal *Bifidobacterium* in improving antitumor immunity and also presents a role in facilitating anti-PD-L1 efficacy; the study was performed on mice. They investigated the formation of melanoma in mice with different commensal microbiota and discovered disparities in spontaneous antitumor immunity that were reduced after cohousing or fecal transfer. This could explain why patients resistant to immunotherapy recovered the response after FMT instillation. The conclusion of this study strengthens the importance of commensal microbes in antitumor immunity and the role of FMT in cancer, especially in patients that undergo immunotherapy [53].

Davar and a group of colleagues conducted a study in 2021 that included 15 patients with advanced melanoma whose condition had progressed or persisted despite taking ICPI. For three months, the researchers gave patients donor-derived FMT through colonoscopy every 14 days, followed by pembrolizumab; 6 out of the 15 patients who were evaluated reacted to the treatment by having their tumors diminish, or their diseases stabilize over time. Additionally, responders displayed a greater abundance of taxa previously linked to immunotherapy response, elevated CD8+ T cell activation, and reduced numbers of interleukin-8-expressing myeloid cells [54].

Baruch and colleagues published a study in 2021 in which he used FMT from 2 allogenic donors who had shown an excellent response to treat 10 patients with ICPI-refractory melanoma (progressive disease during or after ICPI therapy). Three of the ten patients recovered from the immunotherapy response after FMT. According to tumor regression on a PET-CT scan, one of these individuals had a complete response to nivolumab, while the other two had partial responses [55] (Table 2).

Regardless of the limitations of small sample sizes, the fact that two independent cancer centers in different parts of the world with different patient populations reported similar clinical and translational results in accordance with pre-clinical findings using different ICPI (nivolumab versus pembrolizumab) is highly supportive of the validity of these preliminary results [55,56].

Borgers et al. published in 2022 an ongoing double-blinded, randomized phase Ib/II trial in which it will be investigated the safety and efficacy of FMT in anti-PD-1 monoclonal antibodies comparing FMTs derived from ICPI responsive or nonresponding donors in refractory advanced-stage melanoma patients. In this trial, 24 advanced-stage cutaneous melanoma patients who are anti-PD1-refractory will receive an FMT from either an ICPI-responsive or non-responding donor. The investigation will last for almost two years entirely. The primary objective of this trial is the efficacy, defined as a clinical benefit (complete or partial response; durable, stable disease) at 12 weeks and confirmed on a CT scan at 16 weeks, of an FMT treatment in patients with advanced melanoma who were resistant to anti-PD-1 from responsive or non-responding ICPI donors [57].

In order to improve responses and/or reduce toxicity, FMT is increasingly being used to modify gut bacteria in patients receiving cancer immunotherapy [25,46]. In addition to the two studies described, several other clinical trials are investigating FMT along with ICPI in a broader range of tumor types, including renal cell carcinoma, non-small-cell lung cancer, urothelial cancer, and prostate cancer. The majority of these are aimed at patients with similar ICPI resistance and combine FMT with anti-PD-1 monoclonal antibodies [58]. The overall goal of all studies is efficacy and/or safety. The key to developing microbial-based and microbial-targeted therapeutics is the discovery of microbial biomarkers connected with improved responses to cancer treatment [59]. We note that the uniformity of all phases of microbiome investigations, including sampling, storing fecal samples, selecting an experimental design, and using bioinformatics techniques, represents a crucial issue for ascertaining the composition of the microbiota [60].

The new perspectives that FMT opens in recovering response to treatment in melanoma represent an important key discovery in order to improve the survival of oncological patients, especially for patients with melanoma. Additionally, extensive investigations towards interaction between the gut microbiota, immune cells, and clinical response using feces, blood, and tumor samples will increase the effectiveness of FMT therapy in the oncological field.

## 7. FMT and Graft-Versus-Host Disease

Graft-versus-host disease (GvHD) is characterized by inflammation in several organs; stem cell and bone marrow transplants are frequently linked to GvHD. Patients with specific blood illnesses may benefit from allogeneic hematopoietic cell transplantation as a possible curative treatment. Although allogeneic stem cell transplantation can cure many hematological illnesses, GvHD is a leading cause of morbidity and mortality [61]. Systemic administration of high-dose glucocorticoids is the first-line treatment for acute GvHD, but depending on the disease’s severity, only 40–60% of patients react to it [62].

The effectiveness of FMT has been analyzed in the review realized by Maroun Bou Zerdan et al., which compared different studies in patients with corticosteroid refractory GvHD. More than 50% of these patient groups showed responses. More importantly, all of the procedures were well tolerated, with the exception of one patient who experienced lower gastrointestinal hemorrhage and hypoxia and two other patients who experienced bacteremia, all of which were determined to be unrelated to the FMT [3,61,63,64].

Ye Zhao and a group of colleagues conducted a study in 2021, which included 55 patients with steroid-refractory GvHD of the gastrointestinal tract. FMT was administered to 23 patients with grade IV steroid-refractory gastrointestinal-GvHD. Most patients’ microbial richness in terms of variety following FMT was increased when compared to their pre-treatment gut microbiota. Beneficial bacteria, such as *Bacteroidetes* and *Firmicutes*, increased in most patients after FMT [63].

In another pilot study, in eight patients with steroid-refractory gastrointestinal tract GvHD, FMT was administered when immunosuppressive medication and other innovative treatments failed. As a result, all patients experienced reduced clinical symptoms (e.g., decreased stool volume and frequency and less abdominal pain), which could be attributed to the enhanced microbial richness [64].

In a teenager with stage IV GvHD who received four doses of FMT at weekly intervals, Zhang F. et al. looked at the longitudinal dynamics of the gut bacteriome (bacterial microbiome), mycobiome (fungal microbiome), and virome (viral microbiome). After FMT, they revealed inconsistent patterns of modification in the recipient’s gut bacteriome, mycobiome, and virome [65].

Numerous clinical investigations are currently being conducted in numerous institutions across the world as a result of the early achievements shown with FMT in patients with hematologic and oncologic illnesses. Borody et al. reported in 2011 the first “cure” of associated immune thrombocytopenia in a patient receiving FMT for UC [66]. In the study conducted by Goeser et al. on two German tertiary clinical centers, positive outcomes were recorded in patients with GVHD after FMT without observing significant side effects in patients [67].

Nearly 40 studies are now listed on Clinicaltrials.gov. The safety of FMT, its application to prevent and treat GvHD following allogeneic hematopoietic stem cell transplantation, improvement in ICPI response, and management of side effects associated with cancer therapy are the main focus points of these studies. Within the next five years, many of these researchers should disclose their mature data on these results [3,68].

Graft-versus-host disease (GvHD) is a systemic disorder that can be resistant to treatment and may cause irreversible organ failure; in light of this situation, we want to outline the clinical benefits of FMT in reducing symptoms and alleviating the clinical status of the patients, as well as improving the treatment response by restoring healthy microbiota.

## 8. FMT Improves Response to Chemo and Immunotherapy in Colorectal Cancer

Colorectal cancer (CRC) is the second most frequent cancer in women and the third most common cancer in men worldwide. Males have significantly greater incidence and mortality rates than females. Environmental and genetic variables, such as a high-fat diet and disturbed gut flora, can contribute to the development of this illness [69].

One of the most popular treatment plans for colorectal cancer is FOLFOX (5-fluorouracil, leucovorin, oxaliplatin). Chemotherapeutic drugs cytotoxicity can easily lead to unbalanced gut flora, disrupt the gastrointestinal mucosa’s barrier, and mediate the mucosal inflammation known as “gastrointestinal mucositis” [70].

A pilot study was performed in Taiwan to examine the impact and safety of FMT on intestinal mucosal injury caused by 5-FU-based chemotherapy (FOLFOX) in mice with colon cancer. FMT was performed to restore microbial diversity and composition using faces from healthy wild-type donor mice. The results of the study showed that in mice with colon cancer, FOLFOX treatment greatly slowed the growth of tumors. In mice carrying colorectal cancer, FMT reduced FOLFOX-induced severe diarrhea, bacterial translocation, intestinal mucosal damage, and increased long-term survival. FMT also lessened the disturbance of the barrier integrity and intestinal mucosal inflammation brought on by FOLFOX. These findings suggest that FMT may be clinically effective in treating intestinal dysbiosis and toxicity brought on by chemotherapy [70,71,72].

A recent review realized by Sillo et al. describes the importance of the gut microbiome response in case of colorectal cancer; they note the importance of *Proteobacteria Firmicutes* in a pre-clinical study in responders in Anti-CTLA-4 therapy and an abundance of *Bacteroides* in non-responders [73]. Additionally, their review presents another pre-clinical study on mice that outlined that in anti-PD1 therapy, the fecal microbiome in responders was represented by *Akkermansia municiphilia*, *Prevotella* spp., while the non-responders presented more *Bacteroides* spp., *Bacteroides_sp._CAG927* [73,74,75].

This review highlights the importance of a healthy gut microbiome for proper response in the case of immunotherapy and chemotherapy in colorectal cancer. Dysbiosis has been associated with the tumoral process in colorectal neoplasia; furthermore, according to cited studies, FMT may improve gastrointestinal symptoms during chemotherapy and increase the survival rate of the patients [76,77].

## 9. Microbiota Dysregulations in Pancreatic Cancer

An estimated 62,210 individuals are diagnosed with exocrine pancreas cancer each year in the United States, and almost all of them are predicted to pass away from the illness. For both men and women in the United States, pancreatic cancer ranks as the fourth most common cancer-related cause of death. A total of 85% of these tumors are adenocarcinomas that develop from the ductal epithelium. According to data from the 2017 Global Burden of Disease Study and the World Health Organization (WHO) GLOBOCAN database, pancreatic cancer is the seventh highest cause of cancer deaths globally in both men and women [78].

A recent translational pilot study looked at the microbiota changes in pancreatic cancer patients versus healthy volunteers, as well as the impact of FMT from these patients on germ-free mice. The patients were recruited from the Geneva University Hospital. The study included five patients aged 18 years who had been newly diagnosed with pancreatic adenocarcinoma, had no tube feeding or parenteral nutrition, and had not received antibiotic treatment in the previous month. The study also included five healthy volunteers who were matched for gender and age (+/- 5 years) with the included pancreatic cancer patients, were aged 18 years, had a body mass index less than 30 kg/m^2^, had no chronic diseases, and had not received antibiotic treatment in the previous month. The pilot study found that the fecal microbiota of pancreatic cancer patients differed from that of healthy controls. Mice recipients of pancreatic cancer patient feces had lower visceral fat than volunteers but similar immune/inflammatory parameters, and the microbiota of transplanted mice partially reflected the taxonomic composition of bacterial communities of their corresponding human donor. However, research is needed to confirm that feces contain elements associated with metabolic and immune changes [79].

Sammallahti et al. note that *Fusobacterium* in patients with pancreatic cancer was associated with a worse prognosis and concluded that there is a complex relationship between microbiota and tumorigenesis of the pancreas. Moreover, they outline that stool microbiota could be used as a diagnostic and predictive biomarker and, furthermore, could be used as a predictive marker for treatment response [80].

Pancreatic cancer remains a deadly disease despite decades of research with still few effective treatments. We will gain new knowledge and possibly have prospects for the creation of innovative biomarkers and interventional techniques as a result of our increased understanding of the microbiome’s function in pancreatic cancer [80,81,82,83].

Microbiota transplantation may interfere with drug efficacy during cancer therapy, with the ability to either boost or hamper chemo- or immunotherapies. The relationship between diseases, as well as the diagnostic and therapeutic potential of the microbiota, will be evaluated through research combining microbiology, immunology, metabolomics, molecular pathology, tumor genetics, and many dimensions, creating a comprehensive picture of pancreatic cancer genesis and progression.

## 10. Conclusions

The only legally recognized indication for FMT use is the treatment of individuals with recurrent *Clostridoides difficile*-associated colitis, which was the first condition for which it was successfully utilized in modern medicine.

Through all the specialized studies carried out so far, FMT has proven its effectiveness in inflammatory bowel diseases. FMT in IBD could bring hope for patients that present refractory disease or develop resistance to biological therapies such as Infliximab, Adalimumab, or other new molecules, also improving the gut with healthy microbes could reduce inflammation, fistulations, or stenosis, thus improving clinical and endoscopically scores. Moreover, FMT may be a rescue therapy to prevent early surgery.

In patients with neoplasia, the side effects of chemotherapy represent a real problem for patients; FMT is a good and modern method of improving the quality of life of oncological patients and restoring response to treatment in case of resistance. However, due to research limitations, patients with hematologic and oncologic illnesses should only undergo FMT in highly supervised clinical trials.

These clinical benefits, which involve very few adverse effects, should be explored by oncologists to improve patients’ responses to therapies. As presented above in our review, a predominance of good microbes in the gut improves the response to immunotherapy. Additionally, in the case of different types of cancer, this dependence between gut microbiota and tumorigenesis, as well as response to treatment, opens a new perspective on FMT that practitioners should take advantage of. As the cited studies show in the case of colorectal cancer and melanoma, FMT may reduce inflammation, influence cell signaling pathways, reduce tumor cell proliferation, and improve the response to cancer therapy.

Furthermore, with regard to the SARS-CoV-2 virus, we note that it still represents a global health problem. The long-term effects of the disease are not yet known and are being intensively studied. FMT proved its effectiveness in patients who presented gastrointestinal symptoms, but considering that the COVID-19 pandemic is not over, many other studies are needed to concretize the effectiveness of FMT.

Up-to-date, the efficacy of FMT for the treatment of digestive and non-digestive diseases is being developed and improved to fully understand the intestinal microbiota. Nonetheless, the effects proven so far are promising and encourage continuous research in order to establish treatment protocols regarding FMT in various diseases.

## Figures and Tables

**Table 1 pathogens-12-00766-t001:** Studies on FMT clinical benefits in IBD patients.

Kunde, S. et al. (2013) [45]	Included 10 pediatric patients with active UC FMT with enemas for 5 days Clinical remission at 1 week after FMT
Zhang, F. M. et al. (2013) [48]	Study case of CD complicated with fistula Treated with FMT as a rescue therapy
Gianluca Ianiro et al. (2014) [49]	A total of 42 patients received FMT nasogastric or nasojejunal tube and gastroscopy A total of 20 patients received FMT enemas A total of 23 through colonoscopy A total of 11 patients had FMT via gastroscopy and colonoscopy FMT obtained a reduction in symptoms of 71%
Mańkowska-Wierzbicka, D. et al. (2020) [46]	Included 10 patients with active UC Multisession FMT by colonoscopy/gastroscopy FMT improves calprotectin levels and Truelove and Witts scores 16S rRNA gene sequencing was used for metagenomic analysis showing changes and enrichment in the gut microbiota with healthy microbes
Quagliariello, A. et al. (2020) [25]	Pilot study of two pediatric patients with refractory disease FMT showed no adverse effects, and endoscopic appearance was improved
Crothers, J. et al. (2021) [47]	Included 12 patients with active UC Ribosomal 16S bacterial sequencing revealed changes in the microbiota FMT induced clinical remission compared to placebo

**Table 2 pathogens-12-00766-t002:** FMT therapeutic perspectives in melanoma patients.

Youngster et al. 2019 [53]	Five patients with melanoma resistant to therapy FMT via gastroscopy and colonoscopy After treatment, three patients out of five recovered sensitivities to treatment and presented a response to immunotherapy
Davar et al. 2021 [55]	Included 15 patients with advanced melanoma resistant to immunotherapy Patients received FMT every 14 days with pembrolizumab Six patients diminished their tumors/ stabilized their disease
Baruch et al. 2021 [56]	Included 10 patients with refractory melanoma Received FMT from two allogenic donorsThree patients recovered immunotherapy response One patient presented a total response to nivolumab Two patients presented a partial response

## Data Availability

Not applicable.

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
