# Peer review of "Therapeutic Perspectives for Microbiota Transplantation in Digestive Diseases and Neoplasia—A Literature Review"

_pathogens, 2023, doi:10.3390/pathogens12060766_

Round 1

Reviewer 1 Report

A brief summary

In the review titled “Microbiota transplantation from clostridoides difficile to improving immuno-therapy resistance in melanoma- a review of the literature” the Authors intended to describe up-to-date data on the usage of fecal microbiota transplantation (FMT) in the treatment of various diseases i.e. Clostridium difficile infection (CDI), inflammatory bowel diseases (IBD), melanoma, graft-versus-host disease (GvHD), colorectal cancer, pancreatic cancer, COVID-19, and others.

General concept comments

The relevance of the review topic is high as the role of microbiota in the development and treatment of many diseases has been intensively investigated in recent years.  However, the submitted review does not fill the gap in the review literature on this subject. First of all, it lacks completeness in the review topic covered. The Authors only slide over the surface of the subject. None of the described issues on FMT and diseases is appropriately and sufficiently described and backed with original and up-to-date citations. Accordingly, the appropriateness of references is doubtful. The number of references is very low (44) as for the review paper and the majority of them are reviews. Many of the references are inappropriately cited. No reliable conclusions can be drawn from such poor literature analysis. Moreover, the review is written messily and incoherently. It also lacks originality and substantive importance. Doubtful is also its compliance with the scope of the special issue (Clostridium Pathogenesis: Virulence, Host Responses, Microbiome, and Interventions).

Specific comments

1.      The title is misleading and vague. It suggests that melanoma is the main subject of the review.

2.      The first paragraph of the Introduction looks like a copy of reference no 1 with some changes in the order of words and sentences.

3.      In paragraph 2 of the Introduction, the references should be cited after each statement, not at the end of the paragraph.

4.      Section no. 3 – the issues of faces preparation, donor screening, and safety are very interesting, however, the data described is based on unknown references. The cited articles (no. 35 and 36) are inappropriate. The first one is a review paper and the other is an original article describing results obtained on the murine model.

5.      Section no. 4 mixes up “data” on the usage of FMT in various pathologic conditions (CDI, recurrent radiation enteritis,  type 2 diabetes mellitus, obesity, diabetic kidney disease, Parkinson's disease,  Crohn's disease, lupus erythematosus, and rheumatoid arthritis). For the vast majority of the diseases, there is only one citation, predominantly review. For Parkinson’s disease, the Authors cite a protocol for a self-controlled interventional donor-FMT pilot study, which adds no substantive content. Figure 1 is based on unknown data, as the cited reference is incorrect.

6.      Section no. 5 - The coverage of the addressed topic (inflammatory bowel disease and FMT) by this review is extremely poor. There are 472 articles on this subject in PubMed, while the Authors cite 4 references including one case report, two reviews, and one editorial. Moreover, all the citations are incorrect: reference no 25 does not relate to anxiety, depression, and poor quality of life of IBD patients. Ref. no. 26 and no. 27 are probably mixed up. Ref. no. 28 is an editorial about the role of FMT in overcoming immunotherapy-resistant cancers.

7.      Section no. 6 - For the “melanoma and FMT” phrase there are 27 records in the PubMed database, while the Authors cite only six of which four are review papers (one concerning colon cancer) and two references are descriptions of studies on the Medscape webpage. The Authors omitted such important research on this subject as those by Davar et al. (2021, Science; doi: 10.1126/science.abf3363), Borgers JSW et al. (BMC Cancer, 2022; doi: 10.1186/s12885-022-10457-y) or Baruch et al. (Science 2021; 10.1126/science.abb5920). The last one is mentioned, however, it is cited with other, incorrect references.

8.      Section no. 7-10 - The range of addressing the topics addressed in these sections is similar to those described above. Some general statements are mixed up with a relatively detailed description of selected studies. The issues are addressed extremely superficially as there are limited data on up-to-date results. There are too few references and many of them are incorrect.

9.      The conclusions ignore large portions of the literature as the review is based on too few references. Out of 17 000 found, the Authors cited only 44 and most of them are reviews (n=29), one commentary (no. 34), one editorial (no. 28), two articles from the Medscape webpage (no. 29 and 31), and one in the non-English language (no. 14). In my opinion, the proportion between reviews and original articles should be at least reverse and only scientific articles should be cited. As aforementioned, many of the references are inappropriate e.g.:

-          Lines: 70, 97, 105, 186, 200, 233, 278, 287, 306, 327

10.  A large number of careless language errors: grammar and stylistic mistakes.

Minor remarks:

1.      Incorrect URL in ref. no. 18 and no. 20.

The article should be revised by a native speaker or professional language editing service.

Author Response

  1. The title is misleading and vague. It suggests that melanoma is the main subject of the review.

 We thank the distinguished reviewer for this observation. The title has been modified for better clarity.

      The first paragraph of the Introduction looks like a copy of reference no 1 with some changes in the order of words and

We appreciate the distinguished reviewers’ observation in this regard and have corrected and revised the introduction

  1. In paragraph 2 of the Introduction, the references should be cited after each statement, not at the end of the paragraph.

We thank the distinguished reviewer for drawing our attention upon this error. We corrected this issue and cited after each statement

  1. Section no. 3 – the issues of faces preparation, donor screening, and safety are very interesting, however, the data described is based on unknown references. The cited articles (no. 35 and 36) are inappropriate. The first one is a review paper and the other is an original article describing results obtained on the murine model.

We appreciate the thoroughness of the distinguished reviewer’s evaluation.We revised the references and also added some new ways of FMT administration like spores  therapy, liquid capsules etc.

  1. Section no. 4 mixes up “data” on the usage of FMT in various pathologic conditions (CDI, recurrent radiation enteritis,  type 2 diabetes mellitus, obesity, diabetic kidney disease, Parkinson's disease,  Crohn's disease, lupus erythematosus, and rheumatoid arthritis). For the vast majority of the diseases, there is only one citation, predominantly review. For Parkinson’s disease, the Authors cite a protocol for a self-controlled interventional donor-FMT pilot study, which adds no substantive content. Figure 1 is based on unknown data, as the cited reference is incorrect.

We thank the distinguished reviewer for raising this issue in need of clarification. We have included a table to outline new studies and reviews on FMF and revised the references, also added citations for each disease.  

  1. Section no. 5 - The coverage of the addressed topic (inflammatory bowel disease and FMT) by this review is extremely poor. There are 472 articles on this subject in PubMed, while the Authors cite 4 references including one case report, two reviews, and one editorial. Moreover, all the citations are incorrect: reference no 25 does not relate to anxiety, depression, and poor quality of life of IBD patients. Ref. no. 26 and no. 27 are probably mixed up. Ref. no. 28 is an editorial about the role of FMT in overcoming immunotherapy-resistant cancers.

We thank the distinguished reviewer for raising this important question, we added some more studies and also outline the important studies and results through a table.

  1. Section no. 6 - For the “melanoma and FMT” phrase there are 27 records in the PubMed database, while the Authors cite only six of which four are review papers (one concerning colon cancer) and two references are descriptions of studies on the Medscape webpage. The Authors omitted such important research on this subject as those by Davar et al. (2021, Science; doi: 10.1126/science.abf3363), Borgers JSW et al. (BMC Cancer, 2022; doi: 10.1186/s12885-022-10457-y) or Baruch et al. (Science 2021; 10.1126/science.abb5920). The last one is mentioned, however, it is cited with other, incorrect references.

We thank the distinguished reviewer for the provided suggestions. We revised the references, also added some more studies including the suggested ones and revised the citations.

  1. Section no. 7-10 - The range of addressing the topics addressed in these sections is similar to those described above. Some general statements are mixed up with a relatively detailed description of selected studies. The issues are addressed extremely superficially as there are limited data on up-to-date results. There are too few references and many of them are incorrect.

We thank the distinguished reviewer for this concern and recognize the importance of improving the quality of the information, we revised the references and also added some more studies. However,  the current literature is scarce with the studies of FMT in colo-rectal and pancreatic cancer.   

  1. The conclusions ignore large portions of the literature as the review is based on too few references. Out of 17 000 found, the Authors cited only 44 and most of them are reviews (n=29), one commentary (no. 34), one editorial (no. 28), two articles from the Medscape webpage (no. 29 and 31), and one in the non-English language (no. 14). In my opinion, the proportion between reviews and original articles should be at least reverse and only scientific articles should be cited. As aforementioned, many of the references are inappropriate e.g.:

We thank the distinguished reviewer for the points raised. And suggestions, added some more information in the conclusion and revised the language errors. We thank the distinguished reviewer for the offered suggestions in order to improve our paper.

-          Lines: 70, 97, 105, 186, 200, 233, 278, 287, 306, 327

  1. A large number of careless language errors: grammar and stylistic mistakes.

Minor remarks:

  1. Incorrect URL in ref. no. 18 and no. 20.

Reviewer 2 Report

The manuscript of Boicean et al reviews of literature on an emerging field of treating various diseases with FMT. It covers publication on several conditions, starting with well-established treatment of C difficile, other gastro-enteral diseases and various cancers. This review provide sufficient information on what is known on this treatment to justify their conclusion that many more studies are needed to fully understand the intestinal microbiota, especially for non-digestive illnesses.

The manuscript will be improved by addressing these issues:

- The title should be rephrased as IMPROVING … RESISTANCE IN MELANOMA could not be a desirable outcome of the FMT

- Keywords and Acknowledgments sections have no information

- Web links for two references (#5 and #18) do not work

There are numerous occasion of misspelling and some terminology is not used correctly (16 comment are in the attached .pdf file)

Author Response

We highly appreciate the distinguished reviewer’s observation in this regard,  the title has been modified for better clarity, we also added keywords and acknowledgement.

We thank the distinguished reviewer for this concern , we revised the references and also revise the manuscript according to your suggestions. We thank you very much for your valuable opinion.

Reviewer 3 Report

It is necessary to update the FMT research in the world. The comments as below:

1.       However, an international journal needs to display all progress in the world for showing all readers. The current version missed the important developments in: (1) the lab preparation for washed microbiota transplantation; (2) the delivery via colonic transendoscopic enteral tubing; (3) the spores therapy; (4) the government approved new drug in USA and Australia.

2.       The literature review is not enough, the recent review: Encyclopedia of fecal microbiota transplantation: a review of effectiveness in the treatment of 85 diseases. Chin Med J. 2022. Therefore, in Figure 1, I do not think “effectiveness” is suitable to express application of FMT.

3.       I suggest to delete the part "EFFECTS OF FMT IN DISCHARGED COVID-19 PATIENTS", because no trials are ongoing. All of these trials will not have opportunity to move on.

4.       Much more literatures should be reviewed and discussed.

good.

Author Response

 However, an international journal needs to display all progress in the world for showing all readers. The current version missed the important developments in: (1) the lab preparation for washed microbiota transplantation; (2) the delivery via colonic transendoscopic enteral tubing; (3) the spore’s therapy; (4) the government approved new drug in USA and Australia

. We thank the distinguished reviewer for the provided suggestions and appreciation, we added the important development in FMT, we assessed the spore’s therapy, the concept of liquid capsules and the other raised issues.

  1. The literature review is not enough, the recent review: Encyclopedia of fecal microbiota transplantation: a review of effectiveness in the treatment of 85 diseases. Chin Med J. 2022. Therefore, in Figure 1, I do not think “effectiveness” is suitable to express application of FMT.

We thank the distinguished reviewer for this concern and recognize the importance of improving the quality of the information, we revised the references and also added some more studies, including the suggested one, once again we thank you for your suggestions in imporoving our work.

  1. I suggest to delete the part "EFFECTS OF FMT IN DISCHARGED COVID-19 PATIENTS", because no trials are ongoing. All of these trials will not have opportunity to move on.
    We thank the distinguished reviewer for this suggestion, we deleted that part, we appreciate your opinion in improving our work.

  1. Much more literatures should be reviewed and discussed.

We thank the distinguished reviewer for raising this question, we added and discussed more studies on each topic, also outline the important studies through a table on each topic.

Round 2

Reviewer 1 Report

General concept comments

The manuscript has been improved only slightly and still requires substantive and linguistic correction. In my opinion, the review tries to cover too wide a range of diseases. I recommend focusing only on selected diseases or a group of diseases (e.g. bowel diseases or cancer) and making the literature analysis more thorough.

Specific comments

1.      The title still needs correction. It suggests that the transplantation is from Clostridoides difficile.

2.      Introduction: The collective references [3,4,5,6,7,8,9,10] in line 48 are too general to be properly used for supporting statements concerning probiotics and prebiotics and the references in line 54 are incorrect. In the third paragraph, the Authors cite Ge Hong, Li Shizhen, and Eisenman but with no references.

The Authors write: “We note that the literature is scarce with regard to FMT related to extra-digestive diseases in terms of articles and reviews..”, however, there are 508 records on PubMed containing the phrase “FMT and cancer”.

There is no point in placing references at the end of the Introduction (line 94).

3.      Section no. 3. The data presented in lines 130-134 are cited with two references. Which one is correct? Or are the numbers mean values of two studies?

The abbreviations WMT, UC, and CD should be explained at first use.

Reference 14 (line 153) or the conclusion should be revised as they do not go with each other.

The conclusion: “fresh and frozen microbiota transplantation seems to represent the most effective treatment” should be supported by a methodological study.

4.      Section no. 4. The statement in lines 160-162 adds nothing to the content and is out of the scope of the review. The title is uninformative as the vast majority of data presented in this chapter concerns COVID-19. Consequently, the rationale for placing some scarce and superficially analyzed data on FMT in pediatric diseases and others e.g. Parkinson’s disease is doubtful. There are 219 records in PubMed for “fecal microbiota and Parkinson’s disease” while the Authors cite only one which is a protocol for clinical trials (32). If the Authors insist on placing this disease in the review, the literature should be analyzed more thoroughly. The same relates to other diseases. Also, the Authors should explain why they regard FMT as a “controversial treatment” (line 182).

The data placed in Table 1 is too scarce to draw conclusions stated in lines 239-241. Ref. 27 (line 217) does not relate to COVID-19.

Which references in line 276 does the data refer to?

5.      Section no. 5. The cited studies are described too extensively. It would be more valuable to analyze more studies but concentrate on comparing the results.

6.      Section no. (apparently) 6. The title is incorrect. The study by Sivan et al. was done on an animal model. This information is missing. The study by Borgers et al. is not finished so no results are available – the information of such a study should be removed from the Table and placed in the conclusion/discussion of the melanoma chapter. The conclusion of this chapter is weak and does not concern melanoma. Reference 48 (line 411) is incorrect (Medscape).

7.      Section no. (apparently) 7. The literature analysis is too poor e.g. a study by Zhang et al (doi: 10.1038/s41467-020-20240-x.) is missing. Ref. 8 (line 440) and 74 (line 462) are incorrect and ref. 3 (line 445) should be replaced with the original article). There are no conclusions.

8.      Section no. 8. The title is misleading as there are also data concerning immunotherapy in the text. The literature analysis is too poor (only one animal study). The references should be corrected e.g. 56,57 (they are not epidemiological studies). There are no conclusions.

9.      Section no. 9. The same remarks as above. The ref. 38 (line 506) is incorrect.

10.  Section no. 10. The conclusions should be improved.

11.  The Tables are messy.

12.  The text should be extensively corrected by a native speaker or professional linguistic editor. A large number of careless language errors: grammar and stylistic mistakes e.g. “sensibility to immunotherapy”, “to their surprise”, and “medical device” (line 149).

Author Response

  1. The title still needs correction. It suggests that the transplantation is from Clostridoides difficile.

We thank the distinguished reviewer for this observation. The title has been modified for better clarity.

MICROBIOTA TRANSPLANTATION CLINICAL BENEFITS IN DIGESTIVE AND EXTRA-DIGESTIVE DISEASES – A REVIEW OF LITERATURE

  1. Introduction: The collective references [3,4,5,6,7,8,9,10] in line 48 are too general to be properly used for supporting statements concerning probiotics and prebiotics and the references in line 54 are incorrect. In the third paragraph, the Authors cite Ge Hong, Li Shizhen, and Eisenman but with no references.

We thank the distinguished reviewer for drawing our attention upon this error. We have revised those citations and revised the whole paragraph.

The Authors write: “We note that the literature is scarce with regard to FMT related to extra-digestive diseases in terms of articles and reviews..”, however, there are 508 records on PubMed containing the phrase “FMT and cancer”.

There is no point in placing references at the end of the Introduction (line 94).

We appreciate the thoroughness of the distinguished reviewer’s evaluation. We deleted that statement and the reference at the end of the introduction.

  1. Section no. 3. The data presented in lines 130-134 are cited with two references. Which one is correct? Or are the numbers mean values of two studies?

The abbreviations WMT, UC, and CD should be explained at first use.

Reference 14 (line 153) or the conclusion should be revised as they do not go with each other.

The conclusion: “fresh and frozen microbiota transplantation seems to represent the most effective treatment” should be supported by a methodological study.

We thank the distinguished reviewer for raising this issue in need of clarification, we deleted the mean values in order to preserve both references, we explained the abbreviations, and revised the conclusion so it can match the reference and included another study to support our conclusion.

Other effective stool formulations are represented by liqid capsules that infuses  frozen or fresh stool into capsules, lyophilized capsules that are dehydrated, powdered feces and that also showed 78%–87.5% efficacy in CDI and were as effective as fresh/ frozen microbiota transplantation. [13] Youngster, et al. concluded in their study on safety and preliminary efficacy of orally administered lyophilized fecal microbiota product compared with frozen product given by enema that either oral or rectal administration of lyophilized fecal microbiota transplantation showed equivalent efficacy, however they noted that the dose should be higher in lyophilized capsules. [13, 83]

Furthermore, Kao et al. in their study noted, frozen FMT is as efficient as fresh FMT in treating recurrent CDI, also with regard to oral capsules the study noted the same efficiency as FMT delivered by colonoscopy. [84]

  1. Section no. 4. The statement in lines 160-162 adds nothing to the content and is out of the scope of the review. The title is uninformative as the vast majority of data presented in this chapter concerns COVID-19. Consequently, the rationale for placing some scarce and superficially analyzed data on FMT in pediatric diseases and others e.g. Parkinson’s disease is doubtful. There are 219 records in PubMed for “fecal microbiota and Parkinson’s disease” while the Authors cite only one which is a protocol for clinical trials (32). If the Authors insist on placing this disease in the review, the literature should be analyzed more thoroughly. The same relates to other diseases. Also, the Authors hould explain why they regard FMT as a “controversial treatment” (line 182).

We thank the distinguished reviewer for raising this important question., we revised the section and preserved only information about CDI and SARS CoV 2, we also deleted table no 1.  We akso revised ref 27 in line 217.

The data placed in Table 1 is too scarce to draw conclusions stated in lines 239-241. Ref. 27 (line 217) does not relate to COVID-19.

Which references in line 276 does the data refer to?

We thank the distinguished reviewer for this observation, we introduced the regarded references.

In patients with UC and CD, preliminary clinical reports of FMT, shown clinical remission that was maintained over a long period of follow-up in many cases and in a small number of additional cases also documented endoscopic and histologic remission. [38,39,341].

  1. Section no. 5. The cited studies are described too extensively. It would be more valuable to analyze more studies but concentrate on comparing the results.

We highly appreciate the distinguished reviewer’s observation in this regard, we revised some of the results and improved the table to outline the results of the studies.  

  1. Section no. (apparently) 6. The title is incorrect. The study by Sivan et al. was done on an animal model. This information is missing. The study by Borgers et al. is not finished so no results are available – the information of such a study should be removed from the Table and placed in the conclusion/discussion of the melanoma chapter. The conclusion of this chapter is weak and does not concern melanoma. Reference 48 (line 411) is incorrect (Medscape).

We thank the distinguished reviewer for the points raised.We revised the title, also stated that the study of Syvan et al was made on mice, and removed the study of Borgers from the table, we thank you once again for your suggestions in order to improve our work, we revised the conclusion and reference no 48 in line 411.

  1. Section no. (apparently) 7. The literature analysis is too poor e.g. a study by Zhang et al (doi: 10.1038/s41467-020-20240-x.) is missing. Ref. 8 (line 440) and 74 (line 462) are incorrect and ref. 3 (line 445) should be replaced with the original article). There are no conclusions.

We thank the distinguished reviewer for the offered suggestions. We revised the reference missing and also corrected  fer 8 and 74, for reference 3 we also added the original articles and presented a conclusion at the end of the section.  

  1. Section no. 8. The title is misleading as there are also data concerning immunotherapy in the text. The literature analysis is too poor (only one animal study). The references should be corrected e.g. 56,57 (they are not epidemiological studies). There are no conclusions.

We thank the distinguished reviewer for drawing our attention on these aspects.We corrected the title, we corrected references 56, 57 with epidemiological studies cited and also  presented a conclusion of the section.

  1. Section no. 9. The same remarks as above. The ref. 38 (line 506) is incorrect.

We thank the distinguished reviewer for drawing our attention on these aspects.

We corrected the title, we corrected ref 38.

  1. Section no. 10. The conclusions should be improved.

We thank the distinguished reviewer for this observation, we completed the conclusions

  1. The Tables are messy.
  2. We thank you for drawing our attention on this aspect, we improved the tables.

  1. The text should be extensively corrected by a native speaker or professional linguistic editor. A large number of careless language errors: grammar and stylistic mistakes e.g. “sensibility to immunotherapy”, “to their surprise”, and “medical device” (line 149).

We appreciate the distinguished reviewers’ observation in this regard and have corrected the formatting errors throughout the manuscript. English language was revised as well.

Reviewer 3 Report

1.       In general, I suggest to revise this manuscript more rigorous and meticulous.

2.       The cited first author in the test should be added “et al.”or “and colleagues” after the name. It is not study of an author, but all authors.

3.       Line 93-94: The cited references 11\13\14\15\16 should be the original reports on washed microbiota transplantation, spore treatment ( SER 109).

4.       Line 205-213: the study on using capsulized FMT for treating patients with COVID-19 is a rigorous study which might be not reliable, according to the manuscript. Additionally, The whole paragraphs from line 205-217 can be deleted.

5.       Line 218-219: “The first case report of FMT in recurrent radiation enteritis indicated that this therapy was successful in symptom relief”. This is wrong, because the first report on FMT for radiation enteritis should be in Radiother Oncol. in 2020.

6.       Table 1: The listed reports are not all of the recent presentative reports. Therefore, delete this table.

7.       Table 2: To list the studies in the order of years, and show the year after the first author.

8.       What is the conclusion of the present review? The last paragraph must be conclusive to cover the whole review.

No.

Author Response

   In general, I suggest to revise this manuscript more rigorous and meticulous.

We appreciate the distinguished reviewers’ observation in this regard and have corrected the formatting errors throughout the manuscript. English language was revised as well.

  1. The cited first author in the test should be added “et al.”or “and colleagues” after the name. It is not study of an author, but all authors.,

We thank the distinguished reviewer for drawing our attention upon this error. We have revised this issue.

  1. Line 93-94: The cited references 11\13\14\15\16 should be the original reports on washed microbiota transplantation, spore treatment ( SER 109).

We appreciate the thoroughness of the distinguished reviewer’s evaluation, we revised the citations and included the original reports.

  1. Line 205-213: the study on using capsulized FMT for treating patients with COVID-19 is a rigorous study which might be not reliable, according to the manuscript. Additionally, The whole paragraphs from line 205-217 can be deleted.

We thank the distinguished reviewer for raising this important question. We revised the paragraphs.

  1. Line 218-219: “The first case report of FMT in recurrent radiation enteritis indicated that this therapy was successful in symptom relief”. This is wrong, because the first report on FMT for radiation enteritis should be in Radiother Oncol. in 2020.

We thank the distinguished reviewer for the appreciation and provided suggestions. We deleted the information concerning radiation enteritis and we also deleted table no1.

  1. Table 2: To list the studies in the order of years, and show the year after the first author.

We highly appreciate the distinguished reviewer’s observation in this regard. We revised the tables and improved them.

  1. What is the conclusion of the present review? The last paragraph must be conclusive to cover the whole review

. We appreciate the thoroughness of the distinguished reviewer’s evaluation, we added conclusions to each section and also improved the final conclusion.

Round 3

Reviewer 1 Report

The manuscript has been improved in the following aspects: contents of the article, references, and coverage of literature. However, it still requires substantive and linguistic corrections (listed below).

1.      The title still does not sound correct.

2.   The references are still dominated by review articles. This is a review of reviews.

3. Some references seem randomly chosen: e.g. 4 (lines 109), 13 (line 140, 144), 70, 72, 73 (line 383); 77 (line 481 – concerns vaccine), line 223 and 436: which reference concerns the data? Lines 440-443: which reference concerns pre-clinical study? Medscape articles are not acceptable for citing.

4. The text is incoherent. It lacks logical consistency and is difficult to follow its contents and conclusions.

5. Literature analysis for some themes is still too narrow e.g. COVID-19 (only 2 original studies), GvHD and FMT – it lacks some important studies e.g. Zhao et al, 2021, Goeser et al, 2021, Qiao et al, 2023. The statement in the title of paragraph no. 8 is not supported by its literature content. No studies describing FMT usage in pancreatic cancer are cited.

6. The conclusions are messy and not convincing.

7. There are many editorial, stylistic mistakes e.g. line 104 (SARS-CO-2), Table 1 (ref. 39 - the surname of the first author), “To their surprise” (line 318), and others.

8. English still requires extensive correction.

Author Response

  1. The title still does not sound correct.

 We thank the distinguished reviewer for this observation, we revised the title.

  1. The references are still dominated by review articles. This is a review of reviews

We appreciate the distinguished reviewers’ observation in this regard .We included some more studies. .

  1. Some references seem randomly chosen: e.g. 4 (lines 109), 13 (line 140, 144), 70, 72, 73 (line 383); 77 (line 481 – concerns vaccine), line 223 and 436: which reference concerns the data? Lines 440-443: which reference concerns pre-clinical study? Medscape articles are not acceptable for citing.

 We thank the distinguished reviewer for drawing our attention upon this error. We revised all the references.

  1. The text is incoherent. It lacks logical consistency and is difficult to follow its contents and conclusions.

We appreciate the thoroughness of the distinguished reviewer’s evaluation. We have proofread the manuscript, and have corrected all formatting errors throughout it and revised the conclusions.

  1. Literature analysis for some themes is still too narrow e.g. COVID-19 (only 2 original studies), GvHD and FMT – it lacks some important studies e.g. Zhao et al, 2021, Goeser et al, 2021, Qiao et al, 2023. The statement in the title of paragraph no. 8 is not supported by its literature content. No studies describing FMT usage in pancreatic cancer are cited.

We thank the distinguished reviewer for raising this issue, we included the suggested studies and revised the title in section no 8.

  1. The conclusions are messy and not convincing.

We thank the distinguished reviewer for raising this important question, we revised the conclusins.

  1. There are many editorial, stylistic mistakes e.g. line 104 (SARS-CO-2), Table 1 (ref. 39 - the surname of the first author), “To their surprise” (line 318), and others.

We thank the distinguished reviewer for this observation and suggestions, we revised the errors.

  1. English still requires extensive correction.

We thank the distinguished reviewer for the offered suggestions, we revised and corrected the paper.

Reviewer 3 Report

1.       “Clostridoides Difficile” should be used, instead t Clostridium difficile.

2.     Delete the last sentence in the abstract. Also, considering the pandemic years, we focused on the proven effectiveness of FMT in 20 patients with COVID-19.

3.     Are you sure you want to cite “2-85” in Line 43.

4.       The references should be carefully checked. For example: The authorship and pages of the reference 12: “Chen, Y.; Yin, H.; Wang, H.; Marcella, C.; et al.” is wrong. According to the context, this citation is not necessary.

5.   In the methods of FMT, the recent delivery of colonic transendoscopic enteral tubing cannot be missed for showing the whole scope for readers.

6.       Carefully read the manuscript for correcting many minor errors.

Author Response

  1. “Clostridoides Difficile” should be used, instead t Clostridium difficile.

We thank the distinguished reviewer for the points raised, we revised this mistake.

  1. Delete the last sentence in the abstract. “Also, considering the pandemic years, we focused on the proven effectiveness of FMT in 20 patients with COVID-19”.

We thank the distinguished reviewer for the appreciation and provided suggestions, we deleted the sentence.

  1. Are you sure you want to cite “2-85” in Line 43.

We thank the distinguished reviewer for the suggestion, we revised the references.

  1. The references should be carefully checked. For example: The authorship and pages of the reference 12: “Chen, Y.; Yin, H.; Wang, H.; Marcella, C.; et al.” is wrong. According to the context, this citation is not necessary.

We thank the distinguished reviewer for the provided suggestion, we revised the references.

  1. In the methods of FMT, the recent delivery of colonic transendoscopic enteral tubing cannot be missed for showing the whole scope for readers.

We thank the distinguished reviewer for the provided suggestion, we included this important topic.

  1. Carefully read the manuscript for correcting many minor errors.

We thank the distinguished reviewer for the appreciation and provided suggestions, we revised the paper.